# Development of Clause Complexity in Children with Specific Language Impairment/Language Development Disorder: A Longitudinal Study

**DOI:** 10.3390/children10071152

**Published:** 2023-06-30

**Authors:** Claudia Araya, Carmen Julia Coloma, Camilo Quezada, Paula Benavente

**Affiliations:** 1Departamento de Fonoaudiología, Facultad de Medicina, Universidad de Chile, Santiago 8380453, Chile; ccoloma@uchile.cl (C.J.C.); cequezad@uchile.cl (C.Q.); 2Facultad de Pedagogía, Universidad Academia de Humanismo Cristiano, Santiago 7500000, Chile; 3Instituto de Estudios Avanzados en Educación-IE, Universidad de Chile, Santiago 8330014, Chile; 4Escuela de Fonoaudiología, Facultad de Medicina, Universidad de Los Andes, Santiago 8150513, Chile

**Keywords:** grammar, clause complexity, longitudinal study, specific language impairment (SLI), developmental language disorder (DLD)

## Abstract

This paper addresses the grammatical challenges associated with the development of clause complexity, focusing on the performance of a group of monolingual Spanish-speaking schoolchildren with Specific Language Impairment/Developmental Language Disorder (SLI/DLD) in a longitudinal corpus of oral narrative samples. The study examines the presence of interclause relations of subordination and equivalence (hypotaxis and parataxis) in language samples of two groups: an experimental group made up of 24 schoolchildren with SLI/DLD and a control group made up of 24 schoolchildren with typical development (TD). The results show that while both groups use parataxis as the most common relation between clauses in all school grades, there is a significant decrease in paratactic relations and a significant increase in hypotactic relations from first to fourth grade of primary education. Although the development patterns are highly similar, the SLI/DLD group shows greater difficulties in mastering more complex (hypotactic) relations in fourth grade compared to the control group, indicating that it is less sophisticated in the use of these types of complex relations. These findings suggest that focused support on the most complex structures is needed towards the fourth grade of primary education, given the demands of the school academic register from 6 and 7 years of age and the potential problems that the development of clause complexity can cause in school-age children.

## 1. Introduction

Specific Language Disorder (SLI) or Developmental Language Disorder (DLD) is a developmental disorder characterized by a set of difficulties that affect language acquisition [1]. This disorder manifests itself as an important limitation in the expression and/or comprehension of oral language [2], which affects communicative practices in speech and language processing [3,4]. The most frequent deficits in SLI/DLD are related to the morphosyntactic level of language [5,6,7,8,9]. These morphosyntactic difficulties are considered a clinical marker for the diagnosis of the disorder [10]. Morphosyntactic problems of this kind have been observed in several languages [8,11], although most of the information has been obtained from English-speaking children with SLI/DLD [12]. Of particular interest within these morphosyntactic problems are the grammatical challenges related to the development of clause complexity in children experiencing these specific language difficulties, particularly in languages other than English.

### 1.1. Literature Review: Syntactic Complexity in Spanish-Speaking Children with SLI/DLD

Several studies on sentence complexity across different languages indicate that it poses a significant challenge for children with SLI/DLD [13,14,15,16,17,18,19,20]. Research on Spanish-speaking children with SLI/DLD has found that they exhibit difficulties in both sentence comprehension and production, particularly in the area of complex sentences [13,14,21]. Moreover, production difficulties seem to be more pronounced as sentence complexity increases [22]. These challenges are also more prevalent in contexts that require the use of more sophisticated linguistic resources, such as narrative discourse production [23,24].

#### Simple and Complex Structures

Many studies on morphosyntactic complexity in children with SLI/DLD have focused on analyzing their language samples in terms of simple sentences and those related to coordination and subordination mechanisms. It has been found that compared to coordinated sentences, children with SLI/DLD have fewer problems with simple sentences [22]. Furthermore, it has been suggested that these children tend to use simple sentences more frequently than complex ones [25] and prefer using simple and coordinated sentences over complex sentences [14,17,22,26].

The analysis of syntactic complexity in children with SLI/DLD considers different perspectives on the concept of complexity. Some studies include coordinated sentences in their description of compound sentences, which they consider as constitutive of complexity. In this regard, it has been observed that children with SLI/DLD use significantly fewer compound sentences than their typically developing peers [13]. Other studies focus only on subordination, which involves integrating one clause within another [25,27,28]. From this perspective, it has been found that children with SLI/DLD have lower production of subordinate clauses than typically developing children [14,16,21,29,30]. However, some studies have not found these differences to be statistically significant, at least in narrative language samples [25,28].

### 1.2. Longitudinal Development and Trajectory of Syntactic Complexity in Children with Typical Development (TD) and with SLI/DLD

The use of subordinate clauses as a mechanism of syntactic complexity in children’s linguistic development highlights their ability to produce sentences that are dependent on others [31]. Typically developing children begin to use two or more verbs in a sentence around 2 years of age, with complex syntax emerging around 30 months; however, the structures and functions involved in complexity are not fully consolidated until after three years of age [19,28]. The developmental trajectory among preschool children with typical development demonstrates substantial and noteworthy variations. Despite these variations, children demonstrate a certain degree of complexity in their statements before the age of four [32]. However, coordinated clauses remain prevalent during the preschool age [19]. By 6 years of age, they handle syntactically more complex linguistic structures than in preschool years [33], although simple sentences are still used more frequently than complex sentences. In the first years of schooling, from age 6 onwards, the use of subordination increases significantly [34]. Finally, studies indicate that typically developing children use more complex clauses at 10 years than at 8 years [35].

On the other hand, children with SLI/DLD preferably use simple sentences between 4 and 6 years of age [17]. However, certain indicators of complex syntax have also been observed in a longitudinal case study involving a child at the age of 5 years and 9 months [20]. It has also been observed that these children use significantly fewer compound sentences in their narratives between the ages of 4 to 11 [13]. Furthermore, Pavez et al. [36] and Coloma et al. [37] showed that, at 6 years of age, children with SLI/DLD maintained the same level of production of complex structures as a control group of 4-year-olds in narrative samples. Regarding conversation, it has been suggested that both children with SLI/DLD and those with language delay produce fewer complex structures than their typically developing peers [16,38]. In this regard, Hincapié-Henao et al. [21] state that children with SLI/DLD have great difficulty in producing complex verbally formulated structures. Among these structures, constructions that reflect hypotactic relationships with time-related, final, and comparative value are also especially challenging [22].

Describing the development of language longitudinally allows, on the one hand, to determine different growth patterns or trajectories that could define characteristics of typical and atypical language [20,39,40] and, on the other, to investigate in greater depth the issue of the persistence of SLI/DLD difficulties over time [41]. Law and Tomblin [40] mention three hypotheses that explain the possible development patterns of language skills in children with SLI/DLD: (1) they coincide at the same starting point and diverge over time (“deterioration hypothesis”); (2) they develop at the same speed, but stop at a certain point, without further development (“plateau hypothesis”); and (3) they take off later, but their language development, although delayed, parallels that of typical development (“tracking hypothesis”). According to Law and Tomblin [40], a reduced heterogeneity is observed in the growth characteristics of children with language disorders, a trajectory that would be similar to that of children with typical language development, at least in the school years, which would be consistent with the explanation of the tracking hypothesis. This finding coincides with the position that, although these children would be delayed, they would not be qualitatively different from children with TD [42]. In this regard, it has also been suggested that children with TD and SLI/DLD would follow a similar path, although children with SLI/DLD obtain lower results [41]. However, it is necessary to emphasize, at this point, that the competence and speed of development in children with SLI/DLD is lower than in children with typical development with respect to the emergence of syntactic complexity at the beginning of these trajectories [27,37] and that the specific characteristics of this starting point provide evidence that widespread vulnerabilities in complex syntax acquisition could typify SLI/DLD [20].

### 1.3. Approaches for the Description of Syntactic Complexity in Children with TD and SLI/DLD

Various approaches have been utilized to assess complexity in children with typical development and SLI/DLD. These methods offer distinct viewpoints and outcomes, with some focusing on quantitative aspects and others on qualitative aspects. Quantitatively, syntactic complexity has been evaluated based on the average length of specific units, such as sentences, clauses, or utterances. Brown [43] proposed the Mean Length of Utterance (MLU) measurement for analyzing children’s language in early developmental stages, which is systematically related to age and accounts for the development of syntactic maturity [44]. This index has been used in studies on various languages, including Spanish [45], English [46], and Portuguese [47], in children with TD and SLI/DLD. The MLU has been linked to other measures of complexity, enabling researchers to observe that older children produce longer linguistic units containing more clauses [31]. Combining the MLU with other indices of ungrammaticality has facilitated the identification of children with language difficulties [48]. Additionally, measuring the MLU in bilingual children with TD (English–Spanish) has been predictive of their language skills in an English narrative retelling test, although the same result was not obtained in Spanish [49].

An alternative approach to measuring syntactic complexity involves focusing on clauses and their relationships, as proposed by Hunt [50,51] (1965, 1970). According to this generativist perspective, a terminal unit (T-unit) is composed of a main clause and any attached or embedded clauses or non-clausal structures, representing both paratactic (juxtaposed and coordinated clauses) and hypotactic (subordinate clauses) relationships. This method allows for the quantitative growth of T-units to be visualized as a child’s development becomes more complex, with indices increasing in parallel with age, schooling, and intellectual level [51].

Describing language samples of children with SLI/DLD has frequently employed the quantification and structural analysis of simple, coordinated, and subordinate sentences as a means of characterizing language complexity [13,14,21,25,27,28,29]. Within this tradition, sentence complexity is typically interpreted based on the presence or absence of different types of sentences, with complexity often referring to utterances that exceed the limits of a clause [28].

Another approach to measuring language complexity involves a qualitative analysis of language samples, focusing not only on syntax, but also on thematic and discursive criteria. In this tradition, researchers have analyzed the distribution and organization of clauses based on interclausal relationships, which can be classified as isotactic, symmetrical and asymmetrical paratactic, hypotactic, and endotactic. For example, Alfaro, Crespo, and Alvarado [26] studied these relationships in a narrative sample of children with SLI/DLD and typically developing children. They found that the SLI/DLD group produced more paratactic relationships than the TD group, although this difference was not statistically significant. This finding suggests that the SLI/DLD group produced less informative texts, indicating lower complexity.

From a systemic–functional perspective, the interpretation of the role of interdependence between clauses (parataxis and hypotaxis) is different. In this approach, clauses, which are considered as the central unit of grammatical meaning, are capable of expressing different types of meaning simultaneously and their function is explained by how they work together [52]. The logical system of “taxis” (from the Greek: order, arrangement, and category) captures the relationship of dependency and interdependence between adjacent clauses, which may be potentially part of different types of clause complexes [53]. At the semantic level, clause complexes illustrate how a flow of events develops and becomes a text [54]. According to this theoretical perspective, differences in the structure of oral or written texts reflect differences of a semantic nature [52,55].

This research will use the systemic-functional perspective to characterize the development of clause interdependence relationships in a corpus of Spanish-speaking children with SLI/DLD, compared to a group of children with typical development (TD), from a longitudinal perspective. To the best of our knowledge, this type of analysis has not been previously employed to examine this population. Thus, the findings of this study will contribute to providing information on syntactic complexity from a novel perspective. The following research questions will guide this investigation:What is the most frequently used type of clause interdependence in both groups?Do children with SLI/DLD exhibit similar patterns of development in different types of clause interdependence compared to children with TD, from a longitudinal perspective?Are there significant differences in the use of clause interdependence relationships between children with SLI/DLD and TD in each grade, from a cross-sectional perspective?

Our findings indicate that both groups, throughout all grades of primary education, predominantly utilize parataxis as the primary relation between clauses. However, there is a notable decline in the use of paratactic relations and a substantial increase in the use of hypotactic relations from first to fourth grade. Although the developmental patterns are largely similar, the SLI/DLD group faces greater challenges in acquiring more intricate (hypotactic) relations by fourth grade, in contrast to the control group. This suggests that children in this group exhibit less proficiency in employing these complex types of relations.

## 2. Materials and Methods

### 2.1. Participants

The language sample analyzed in this study comprised 144 oral narrative texts, which were produced by a total of 48 children, including 24 children diagnosed with SLI/DLD (10 girls and 14 boys) and 24 typically developing (TD) children (10 girls and 14 boys). Participants’ demographic information is shown in Table 1. Both groups of children were recruited for the elicitation of narrative productions on three occasions: when they were in first, second, and fourth grade, respectively, in primary schools that had a similar performance level in the Chilean national assessment of learning outcomes (SIMCE). Due to financial and administrative difficulties, it was not possible to obtain language samples in the third grade.

Before the study, the parents, caregivers, or legal guardians of all participating children provided informed consent for their children to participate in the study, and the research proposal and consent letter were approved by the Ethics Committee of the School of Medicine of the University of Chile.

#### Selection Criteria

The participants with SLI/DLD in this study were selected from Chilean schools with Integration Programs designed to facilitate the inclusion of students with special educational needs in regular schools. All these children had previously been diagnosed with SLI/DLD by speech therapy specialists during their pre-school education, around the age of five. The diagnosis was made following the guidelines established by the Chilean Ministry of Education (MINEDUC) at the time of the study [56], which require the use of at least two instruments with national reference standards to evaluate language proficiency: one for comprehensive and one for expressive skills. In this research, the participants with SLI/DLD were evaluated using the Screening Test of Spanish Grammar (STSG) [57], which includes a subtest to assess morphosyntactic abilities at both levels required by MINEDUC. Children were selected based on their deficient performance in this subtest.

The Screening Test of Spanish Grammar [57] consists of two subtests: expressive and comprehensive, with 23 items each. The grammatical aspects evaluated by both subtests include different types of sentences (affirmative, negative, and passive), pronouns (personal, indefinite, demonstrative, relative, and interrogative), verbs (person, number, tense, periphrastic, and copulative verbs), and possessive adjectives. The grammatical aspects assessed by the Screening Test of Spanish Grammar coincide with those that pose the most significant grammatical challenges for children with Spanish-speaking SLI/DLD, namely, function words and verbs [58,59,60], in addition to sentence grammaticality-related aspects [25].

The selection process for participants with SLI/DLD in this study was based on their scores on the Screening Test of Spanish Grammar. Scores lower than 26 on the expressive test and 35 on the receptive subtest were considered deficient. The control group was selected from among the classmates of the SLI/DLD group who showed adequate academic performance for their educational level and did not present language or learning difficulties, as reported by their teachers.

After forming both groups, their nonverbal cognitive abilities and auditory performance were assessed to rule out any aspects that could influence the diagnosis of SLI/DLD. Nonverbal cognitive abilities were evaluated using the Raven Colored Progressive Matrix Scale, which can be administered regardless of educational level or linguistic abilities. All children in this study scored above the 25th percentile, which is considered within the normal range for this test [61]. Hearing was assessed using a swept frequency audiometer (Interacoustics, model AD629) at frequencies of 500, 1000, 2000, and 4000 Hz. Normal hearing for children in both groups was defined according to international criteria proposed by the World Health Organization [62], with the detection of an average intensity of less than 20 dB HL at frequencies 0.5, 1, 2, and 4 kHz.

Table 2 shows a detail of the cut-off criteria used to diagnose the participants with SLI/DLD in this study. 

### 2.2. Procedure

#### 2.2.1. Corpus Elicitation and Transcription

Individual evaluations were conducted in a private room, and each recording session lasted an average of 20 min. The Narrative Discourse Assessment protocol [63] was utilized to obtain language samples. Trained speech therapists instructed the participants to listen to three stories, which were read aloud without the support of images. After hearing each story, the participants were asked to retell them. The recorded material was transcribed according to the protocol’s instructions. Stories were chosen as the material to work with because children with SLI/DLD tend to produce more words per clause when telling stories than in spontaneous conversation. Furthermore, narratives encourage the use of complex syntactic structures in both children with SLI/DLD and typically developing children [25,34]. Additionally, retelling was chosen without the support of images because their presence might encourage the summative chaining of clauses for coordination [64].

#### 2.2.2. Data Analysis

The language samples collected were divided into clauses and organized into a matrix to analyze each clause based on the ideational metafunction of the Systemic Functional Grammar in its logical component. This allowed the relationships between clauses in the language sample to be identified. The data were tabulated using the taxis system, which is a system that observes the grammatical articulation of clauses in degrees of interdependence [52].

The corpus clauses were categorized based on their relationship with other clauses within the same clause complex. Simplex type clauses, which are not formally related to any other clause, were distinguished from clauses related to hypotaxis (dominance and dependence) and parataxis (equivalence relation). A subtype of paratactic relationship was also identified, which is commonly used in oral narrative texts, in which a series of events with time-related value are integrated within the same sub-sequence. This type of parataxis, typically expressed in the corpus studied through the conjunction “and”, was labeled as “time-related parataxis”. Table 3 presents an example of clause segmentation and analysis of the taxis system used in this study.

The analysis was conducted by two authors of the study. After the completion of the first round of analysis by one author, the second author reanalyzed 100% of the data. The agreement for identification, segmentation of clauses, and count was 98.8% and 92% for the types of taxis. Discrepancies were resolved by consensus. Once the clauses were labeled, three values were calculated, expressing the number of times each participant used the three types of clauses of interest as well as the total number of clauses produced. These absolute frequencies were then converted to a percentage relative to the total number of clauses produced for each student. Therefore, the measures analyzed represent the percentage of each type of clause with respect to the total number of clauses produced, allowing for the normalization of values for each participant and removal of potential biases caused by varying lengths of their oral productions.

The collected data were analyzed and visualized using the statistical software R (version 4.3.0) [65,66]. As the primary goal of the study was to illustrate the differences between the groups rather than establish that such differences exist, a 2 × 3 mixed factorial ANOVA was not employed to explore main effects or interactions. First, the percentages of the three types of clauses used were compared within each group and each course. Then, the same comparisons were made to observe the differences in production percentages between the three selected courses for each group and for each type of clause. As both course and clause type corresponded to intra-subject variables, mixed-effects models were used for both analyses, with participants as a random factor and “clause type” and “course” as fixed-effect factors, implementing contrast and Tukey’s post-hoc test in both cases. Finally, the performance of both groups was compared for each type of clause and for each year using Wilcoxon tests.

## 3. Results

Table 4 presents detailed descriptions of all measures for both groups. Figure 1 complements the table by displaying the means and confidence intervals for each measure, along with the results of the Tukey post-hoc contrasts at each level, including the corresponding groupings of means.

In relation to the first question of the study, the results indicate that time-related parataxis is the most commonly used resource by both groups, albeit with some differences. Among children with typical development (TD), all three clause types differ significantly from each other in first and second grade, while by fourth grade, the differences between time-related parataxis and hypotaxis are no longer significant. Among children with specific language impairment/developmental language disorder (SLI/DLD), the three types of clauses are significantly different from each other in second and fourth grade, but not in first grade, where the production of hypotactic and paratactic relations does not differ significantly. Additional details on the mean differences and their grouping can be found in Appendix A and Appendix B.

Regarding the second question of the research, post-hoc contrasts revealed that the evolutionary pattern of time-related paratactic relationships is highly similar between the two groups. In both cases, the use of these relationships decreases significantly from a high level in first grade to a lower level in fourth grade (with a slightly more marked decrease in the case of children with TD). For hypotactic relationships, identical mean clusters were observed for both groups. However, in contrast to the findings for time-related paratactic relationships, the percentage of production significantly increased between first and fourth grade. Finally, no substantial differences were observed between the groups in terms of paratactic relationships, with a low percentage of production in both groups and a slight difference in second grade.

Figure 2 displays the means of both groups across each school grade, providing a breakdown of the results according to the three types of interclausal relationships.

In relation to the third research question, the results of the Wilcoxon tests indicated that there were statistically significant differences between the SLI/DLD and TD groups in terms of time-related paratactic relationships in both second (W = 165.5, *p* = 0.01) and fourth grade (W = 185.5, *p* = 0.04). Significant differences in the use of paratactic relations were only observed in second grade (W = 435.5, *p* < 0.001), while significant differences in hypotactic relationships were only observed in fourth grade (W = 386, *p* = 0.04).

Overall, the results indicate that the relative importance of clause types is similar between the two groups. Specifically, paratactic relationships exhibit little change across grade levels, with mean percentages of usage remaining low (around 15% for most grades and slightly above 20% for only one grade) in both groups. However, significant differences were observed between the two groups in the changes observed throughout the school grades in the percentages of production of time-related paratactic relationships and hypotactic relationships. In both groups, significant changes were observed in the percentages of production between first and fourth grade, indicating an increase in the production of more complex relationships and less use of simpler relationships. Nonetheless, children with SLI/DLD showed greater difficulties in mastering more complex relationships compared to TD children, as significant differences were shown for both types of clauses in fourth grade. Therefore, the results suggest that although the developmental patterns are highly similar, the performance of children with SLI/DLD does not reach the same level of sophistication as children with TD in fourth grade.

## 4. Discussion

In relation to the first question of this study, which concerns the type of clause interdependence relationship that occurs most frequently in each group, a clear predominance of paratactic relationships can be observed among clauses related in a time-related manner through the conjunction “and”. This predominance begins to diminish as hypotactic relationships increase towards the fourth grade in both groups, although parataxis remains the type of relationship with the greatest presence in the examined corpus. The high presence of the conjunction “and” in children’s speech can be attributed to its functionality as a predominant coordination link, which serves to express various types of relationships [67]. This feature could account for its frequent use in the analyzed corpora, particularly in language samples of children with SLI/DLD. Alternatively, the greater frequency of simple structures coordinated by “and” could be attributed to a semantic mechanism used by children with SLI/DLD to compensate for their syntactic deficit, as suggested by van der Lely and Marshall [68].

However, since both groups show the same pattern of use, it is important to question how much of this result is directly linked to the content of the stories used and, as posed by Winters et al. [69], how specific narrative assessment measures can result in greater group differences. Alarcón and Auzá [70] obtained opposite results to those observed in the present study, finding a greater use of subordination than coordination in a recounting task among first grade children with TD. They observed that the original story of their task favored the explication of multiple conditional and causal relationships, expressed through relationships of dominance and syntactic dependence, which could have had an impact on their results. Nevertheless, most studies with a structural focus coincide with the results of our study, in that coordination relationships and simple syntax are more commonly observed in narrative corpora, particularly in narrative samples of children with SLI/DLD [13,16,17,21,38].

In relation to the second question of this study, it was possible to corroborate that children with SLI/DLD and TD follow a similar developmental pattern in the different types of clause interdependence relationships from first to fourth grade, although the results of the SLI/DLD group remain below those of the control group. This finding is significant, as there is insufficient information about how the grammar problems of these children evolve over time [27]. It should be noted that one of the main characteristics of this disorder is the persistence over time of difficulty in acquiring and developing oral language [71]. Although the distinction between disorder and delay is yet to have empirical evidence [2], the persistence of performance below the levels reached by the control group in this study seems to indicate a language development disorder [1], despite the progress observed in all aspects analyzed.

Through a longitudinal analysis of the data from both groups, two key aspects have been identified: (1) the similarity in the pattern of development between the SLI/DLD and control groups, and (2) the within-group advances that enable the detection of significant changes from one school grade to the next. In relation to the first aspect, previous research has noted that the development pattern of the SLI/DLD group is similar to that of the control group, although there is a gap in each of the measures analyzed. This finding is consistent with other studies that have reported lower competence and slower development in children with SLI/DLD compared to typically developing children [37,41,72]. The “tracking hypothesis” proposed by Law, Tomblin, and Zhang [40] posits that children with SLI/DLD may experience delayed language development compared to their typically developing peers, but that their language development follows a similar trajectory, at least during the school years. This finding aligns with the notion that, although children with SLI/DLD may be delayed in language acquisition, they are not qualitatively different from typically developing children [42], which is also supported by Leonard [8], who suggests that their acquisition process is part of normal language development.

Secondly, the longitudinal view allows us to observe significant changes from one school grade to the next, particularly in relation to time-related paratactic and hypotactic relationships. The results indicate that there is significantly less use of time-related parataxis and significantly more use of hypotaxis towards fourth grade within each group. However, a statistically significant difference is observed in the TD group’s transition from second to fourth grade, which is different from the behavior of the SLI/DLD group. These results are consistent with previous research, which suggests that children with SLI/DLD demonstrate an early emergence of some aspects of syntactic complexity, but their performance in terms of competence is lower compared to control groups, and their rate of development is different [72]. These different patterns of change, which vary depending on the dimension being considered, have also been found in other longitudinal studies of children with SLI/DLD [73].

On the other hand, the explosive increase in hypotactic relations in children with TD from second to fourth grade is related to their increasing capacity to produce complex and diversified sentences with age during childhood [31,35]. The schooling process also poses communicative challenges that have a significant impact on the development and consolidation of new hypotactic complex forms, such as final and concessive subordinate structures, which require mastery of the subjunctive [19] (Serra et al., 2000).

In summary, this longitudinal intra-group perspective enables us to appreciate more subtle changes and differences, such as those that occur with SLI/DLD children in their transition from second to fourth grade.

On the other hand, when considering the possibility of significant differences between groups in each school grade, the most relevant results reveal two interesting findings: (1) both groups behave similarly in first grade according to the measures analyzed, and (2) by fourth grade, the SLI/DLD group tends to use significantly more simple interclausal relationships (such as time-related parataxis), while the TD group tends to use more complex interclausal relationships (such as hypotaxis). On the other hand, a specific pattern emerges in second grade. While the SLI/DLD group shows similar hypotactic relationships compared to the TD group, they differ in their use of simpler relationships. It has been suggested that by this age, typically developing children fully develop their grammatical skills [19]. A longitudinal study with a SLI/DLD group found similar developmental patterns at age 7 [40]. Between first and second grade, significant advances in grammatical skills are observed, particularly in the length and complexity of sentence production [31,74].

## 5. Conclusions

Considering the evidence presented, it can be concluded that the SLI/DLD group exhibits a developmental pattern similar to that of the control group in all aspects analyzed across their transition from first, second, and fourth grades (longitudinal perspective), but a distinct one when comparing the groups with each other in each school grade. The most significant result from this latter perspective pertains to the SLI/DLD group’s tendency, in fourth grade, to continue using simpler interclausal relationships, as compared to the TD group, which shows significantly more complexity in the way it interrelates clauses at the same school grade.

The problems that clausal complexity can cause in school-age children can be related, from a functional perspective, to the academic demands that they must comprehend and produce when they enter school [75]. The increasing complexity demands of the school curriculum at 6 and 7 years of age determine the use of increasingly complex grammatical structures [70]. When comprehension problems arise in the classroom, it is common to underestimate the complexity of academic language used with children in this setting and to associate these difficulties solely with the content of the classes rather than the way in which they are taught [76]. In this sense, it is not possible to isolate language learning from all other aspects of learning [77]. The complexity that characterizes the language that children are expected to use in school includes many structural levels at the word, sentence, and text levels [78]. Therefore, children need to learn to process these extremely complex structures to function successfully in school. This is crucial since it is known that many children with SLI/DLD drag those linguistic difficulties into adolescence and even adulthood when they are not adequately addressed in early clinical and educational contexts [79].

In summary, the findings presented in this study regarding the development of clause complexity in Spanish have important implications for educational practice, particularly considering that grammatical abilities play a crucial role as foundational tools for various challenging school activities faced by children with SLI/DLD. This longitudinal study provides valuable and novel insights into the developmental trajectory of children with SLI/DLD compared to typically developing children in their utilization of more complex structures, specifically hypotactic relations. It is observed that children with SLI/DLD exhibit a similar developmental pattern to their TD peers in the use of these structures; however, they consistently perform significantly below the control group, particularly by fourth grade.

Based on these findings, it is crucial for grammatical interventions in educational settings to consider two key aspects. Firstly, during first grade, children with and without SLI/DLD exhibit similar usage of interclausal relations; however, this similarity does not guarantee subsequent adequate performance. Therefore, educational support at this stage should ensure an equal level of content for all students while also providing targeted preventive attention to children with SLI/DLD, even if their performance does not yet significantly differ from that of their TD classmates. This support could be implemented through the utilization of more implicit teaching techniques, as metalinguistic abilities are still underdeveloped at this age. Frizelle et al. [80] illustrate a wide range of language intervention methods that could prove beneficial in educational environments as well.

Secondly, given the significant differences observed by fourth grade, it is essential to address the challenge faced in both educational and clinical practice: providing specialized support to address these difficulties before children with SLI/DLD reach this stage. These findings also raise questions about the effectiveness of current support offered to students with DLD during the initial years of schooling and, more importantly, challenge the prevailing theoretical notion of SLI/DLD as a temporary condition, which in some countries limits institutional support beyond the age of 9.

In conclusion, it is important to acknowledge the limitations of this study. Firstly, due to financial and administrative constraints, we were unable to obtain language samples in the third grade, which limits the comprehensive nature of our findings. Secondly, there are challenges and limitations regarding the generalizability of our results across diverse settings. These factors hinder our ability to provide universally applicable conclusions. Nonetheless, despite these limitations, this study provides valuable insights within its specific parameters, highlighting the potential for further inquiry to enhance the understanding of applicability in diverse linguistic and cultural contexts.

## Figures and Tables

**Figure 1 children-10-01152-f001:**
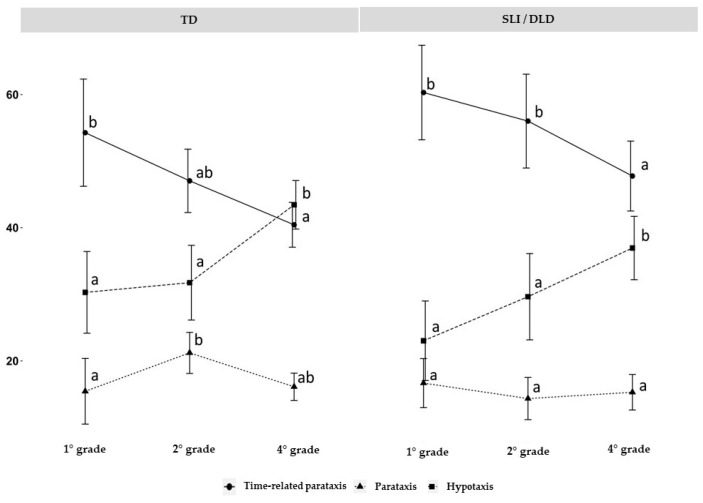
Graph of means and percentages of use of interclausal relations in each school level, separated by group. Error bars represent 95% confidence intervals. The letters indicate the groupings of means according to Tukey’s post-hoc analysis.

**Figure 2 children-10-01152-f002:**
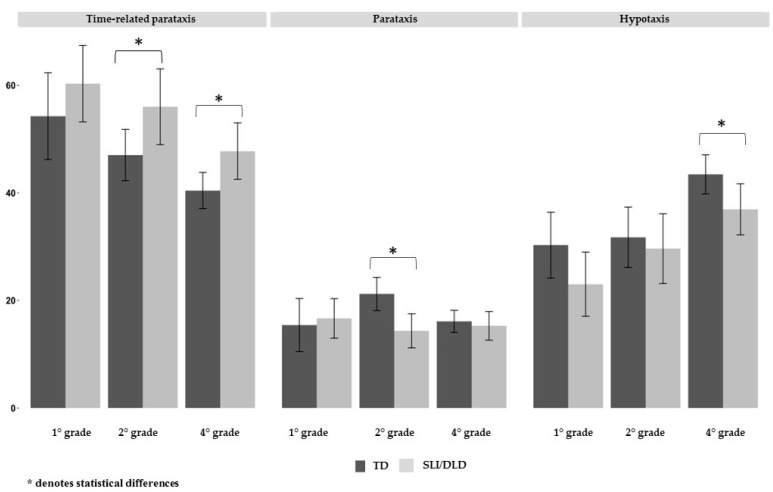
Means of the SLI/DLD and TD group in each school grade and by type of taxis.

**Table 1 children-10-01152-t001:** Participants’ demographic information.

	AGE			GENDER		SCHOOL TYPE *	
	1°	2°	4°	Female	Male	Public	Private
SLI/DLD	6.7	7.7	9.7	10	14	21	3
TD	6.5	7.5	9.5	10	14	21	3

* Public schools in Chile are funded and administered by the government or local authorities.

**Table 2 children-10-01152-t002:** Instruments and cut-off scores used to identify children with SLI/DLD.

Instruments	Normal	At Risk	Deficit
STSG (expressive) *	>36	26–36	<26
STSG (receptive)	>41	38–41	<35
Raven	>15		
Audiometry test	<20 dB		

* STSG (Screening Test of Spanish Grammar, expressive and receptive); Raven (Raven Colored Progressive Matrices Scale). The figures in the table indicate the cut-off scores and the range of normality, risk, and deficit, according to the scale of each instrument.

**Table 3 children-10-01152-t003:** Example of clause segmentation and analysis of the taxis system.

Clauses	Types of Taxis	Clause Complex
	Simplex	
	Hypotaxis	
	Parataxis	
	Time-Related Parataxis	
*La ardillita solo miraba por la ventana. ** (The little squirrel just looked out the window.)	Simplex	1
*No podía salir de su casita ese día* (She couldn’t leave his house that day)		2
*ni jugar con los amiguitos* (nor play with friends)	Parataxis	
*porque estaba muy gorda, gorda* (because she was very fat, fat)	Hypotaxis	
*y se puso muy triste* (and she became very sad)	Time-related parataxis	
*porque no podía salir* (because she couldn’t go out)	Hypotaxis	
*y ahí la llamaban los animalitos* (and the little animals called her)	Time-related parataxis	
*y ella no podía salir y eso.* (and she couldn’t get out and so).	Time-related parataxis	

* Original Spanish language samples were translated into English (Source: Authors’ own elaboration).

**Table 4 children-10-01152-t004:** Descriptions of all measures observed for both groups.

Grade	Taxis	Group	Measure	SD
1st Grade	TR parataxis *	SLI/DLD	54.27	19.07
		TD	60.3	16.82
	Parataxis	SLI/DLD	15.44	11.68
		TD	16.67	8.72
	Hypotaxis	SLI/DLD	30.29	14.52
		TD	23.03	14.13
2nd Grade	TR parataxis	SLI/DLD	47.04	11.27
		TD	56.01	16.68
	Parataxis	SLI/DLD	21.21	7.3
		TD	14.35	7.5
	Hypotaxis	SLI/DLD	31.75	13.29
		TD	29.64	15.34
4th Grade	TR parataxis	SLI/DLD	40.44	7.96
		TD	47.76	12.43
	Parataxis	SLI/DLD	16.12	4.85
		TD	15.29	6.31
	Hypotaxis	SLI/DLD	43.44	8.62
		TD	36.95	11.26

* TR stands for time-related parataxis.

## Data Availability

The data presented in this study are available on request from the corresponding author. The data are not publicly available due to privacy concerns. It is essential to protect the confidentiality and privacy of the participants, because we are dealing with sensitive information such as language impairments or disorders.

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
