# Peer review of "Development of Clause Complexity in Children with Specific Language Impairment/Language Development Disorder: A Longitudinal Study"

_children, 2023, doi:10.3390/children10071152_

Round 1

Reviewer 1 Report

I appreciate the focus on this topic, and very good theoretical background presented in the form of a developmental perspective. However, it should be strongly emphasized that there are huge individual differences in a developmental trajectory in children population of preschool age (as seen in some places of the literature discussion). We can find a nice summarizing of different approaches to measuring language complexity in this paper, and I see a wide range of the relevant and current resources used for this purpose (although, the are a few very old books found in references as well.. .!).

Hopefully, the authors are aware of the difficulties in reaching a widely valid results in such a research (in an international context) due to the national language differences and specifics. On the other hand, methodology and research design have been presented adequately, and the complex tool set was used to check all particular aspects of communication abilities of children... The ethical relevance was not ignored her as well.

As for the research results, they were not surprising but rather expected - anyway, they were proven in a good manner. The authors were aware of risks and challenges hidden in the results described and presented in this paper, too, and it seems that the conclusions can be very useful and appreciated even for the educational (school) and clinical approach children with SLI/DLD.

Author Response

Reviewer 1:

We have carefully considered your suggestions and have made the following changes in the revised version:

  • We acknowledge Reviewer 1's comment about the wide range of individual differences in the developmental trajectory of children in the preschool age population. In the literature discussion, we have now emphasized the significant individual differences and variations seen in the development of language complexity in this specific population (lines 77-78).
  • Regarding the references, we appreciate Reviewer 1 for flagging the inclusion of older books. After careful review, we have considered removing some of them. However, we have concluded that these older books are essential for establishing the fundamental principles of measuring language complexity. Therefore, we believe it is necessary to keep them in the references.
  • We acknowledge Reviewer 1's comment on the challenges in achieving widely valid results in such research, especially in an international context due to language differences and specifics. We have addressed the limitations and challenges related to the generalizability of our findings in different linguistic and cultural contexts within the Conclusions section.

Reviewer 2 Report

We appreciate that it is an interesting study and a good contribution to the literature.

We suggest to the authors:

- to mention the source of each table/figure, even in the case that is the author's elaboration

- to highlight the limits of the research.

Author Response

We have carefully considered your suggestions and have made the following changes in the revised version:

  • We appreciate Reviewer 2's positive feedback regarding the study's interesting nature and contribution to the literature.
  • In response to Reviewer 2's suggestion, we have ensured that the source of each table/figure is clearly mentioned in the revised manuscript, even in cases where it is our own elaboration.
  • Additionally, we have highlighted the limitations of the research, as suggested by Reviewer 2, to provide a comprehensive and transparent account of the study's scope in the Conclusions section.

Reviewer 3 Report

I have a few recommendations and comments.

1. The abstract is very clear meeting the requirement for a scientific structure. 

2. In the Introduction, I noticed you have [5,6,7,8,9]. This should be [5-9]. Check the journal guidelines and make sure you make the corrections in the entire paper

3. The introduction should be kept simpler and shorter. It should be just an intro to the topic. The authors should add a Literature Review section after the Introduction comprising the information they already have but is wrongfully put in the Introduction. 

4. The method, the procedure are very well explained, as well as the results of the study conducted by the authors. 

5. The Discussion includes exactly what it should, a presentation of results in relation to other studies. 

6. Still, Discussion and Conclusions are not the same thing, so, the authors should include a last section, Conclusions, where they should present Theoretical and Practical implications of their study (focusing on the usefulness, novelty and originality of the paper), Limitations of the study (they were in the Methods mentioned but they should be reiterated here) and Future Research Directions. The last part of Discussions already has some of this information but it should be moved to Conclusions and also added what is missing. 

7. The authors should add in the Literature Review, Discussion or Conclusions (where they consider it appropriate) more recent references which I noticed they do not have. At least a few between 2021-2023. 

Author Response

We have carefully considered your suggestions and have made the following changes in the revised version:

  1. Abstract: We appreciate Reviewer 3's positive feedback on the clarity of the abstract.
  2. Introduction: We have addressed Reviewer 3's comment about the citation format. As per the journal guidelines, we have corrected the citation style from [5,6,7,8,9] to [5-9] throughout the paper.
  3. Introduction and Literature Review: In response to Reviewer 3's suggestion, we have restructured the manuscript. The Introduction now focuses solely on introducing the topic. We have created a separate section called "Literature Review" after the Introduction to house the relevant information previously included in the Introduction.
  4. Method and Results: We are grateful for Reviewer 3's positive feedback regarding the clarity of the method and procedure sections, as well as the presentation of results. No further changes were made in these sections.
  5. Discussion: We appreciate Reviewer 3's acknowledgment that the Discussion adequately presents the results in relation to other studies. No further changes were made in this section.
  6. Discussion and Conclusions: We have carefully considered Reviewer 3's comments regarding the distinction between the Discussion and Conclusions sections. In response, we have made the following revisions:
    • We have added a separate section titled "Conclusions" at the end of the manuscript.
    • In the Conclusions section, we have included a discussion on the theoretical and practical implications of the study, highlighting its usefulness, novelty, and originality.
    • We have reiterated the limitations of the study in the Conclusions section, as suggested by Reviewer 3.
    • We have also included some "future research directions" in the Conclusions section.
  7. Recent References: We have considered Reviewer 3's suggestion to include more recent references and added some new relevant references published between 2021-2023 to enhance the currency of the manuscript. We have highlighted them in the References section.